# K-Clique Multiomics Framework: A Novel Protocol to Decipher the Role of Gut Microbiota Communities in Nutritional Intervention Trials

**DOI:** 10.3390/metabo12080736

**Published:** 2022-08-10

**Authors:** Carlo Mengucci, Lorenzo Nissen, Gianfranco Picone, Corinne Malpuech-Brugère, Caroline Orfila, Luigi Ricciardiello, Alessandra Bordoni, Francesco Capozzi, Andrea Gianotti

**Affiliations:** 1Department of Agricultural and Food Sciences (DISTAL), University of Bologna, 47521 Cesena, Italy; 2Interdepartmental Centre for Agri-Food Industrial Research (CIRI Agrifood), University of Bologna, 47521 Cesena, Italy; 3Unité de Nutrition Humaine (UNH), Université Clermont Auvergne, INRAE, CRNH Auvergne, F-63000 Clermont-Ferrand, France; 4School of Food Science and Nutrition, University of Leeds, Leeds LS2 9JT, UK; 5Gastroenterological Unit, Department of Medical and Surgical Sciences (DIMEC), University of Bologna, 40138 Bologna, Italy

**Keywords:** faecal metabolomics, metagenomics, volatilome, machine learning, network of interactions, k-clique communities

## Abstract

The availability of omics data providing information from different layers of complex biological processes that link nutrition to human health would benefit from the development of integrated approaches combining holistically individual omics data, including those associated with the microbiota that impacts the metabolisation and bioavailability of food components. Microbiota must be considered as a set of populations of interconnected consortia, with compensatory capacities to adapt to different nutritional intake. To study the consortium nature of the microbiome, we must rely on specially designed data analysis tools. The purpose of this work is to propose the construction of a general correlation network-based explorative tool, suitable for nutritional clinical trials, by integrating omics data from faecal microbial taxa, stool metabolome (1H NMR spectra) and GC-MS for stool volatilome. The presented approach exploits a descriptive paradigm necessary for a true multiomics integration of data, which is a powerful tool to investigate the complex physiological effects of nutritional interventions.

## 1. Introduction

In recent years, high-throughput techniques and availability of multiomics data have revolutionised the study of complex biological processes that link nutrients, bioactive compounds, food, and the whole diet to human health. On one side, variations occurring at multiple levels, such as the genome, epigenome, transcriptome, proteome, and metabolome levels, alongside clinical information after nutritional clinical intervention provide useful insights to better understand the aetiopathogenesis of diseases and to formulate a nutritional strategy to reduce disease risk. On the other side, the availability of omics data providing information from different layers highlights the need to develop integrated approaches combining individual omics data, in a sequential or simultaneous manner.

Tools and methods that perform the integration of multiple omics data generated from the human genome, proteome, transcriptome, metabolome, and epigenome have been recently reviewed [1,2], among others. This complex scenario is further complicated when the relationships linking diet/foods and human beings are considered alongside the role of the microbiota.

Nowadays, the reciprocal interaction between diet/food/food components and the gut microbiota is scientifically evident. A recent bioinformatics study using a machine learning framework (based on the scikit-learn Python package) screened 1203 different gut microbiomes and found many significant associations between a group of microbes and specific nutrients, foods, and food groups [3]. The correlation between diet and the modulation of gut microbiota has been scientifically strengthened [4,5], as has the impact of the microbiota on the metabolisation and bioavailability of food components [6,7].

It is now clear that evaluation of the diet–microbiota interaction cannot be ignored when deciphering systematically and holistically the effects of the diet/food/food components on human health. The nature of the microbiota cannot be described by a simple set of differently abundant microbes, but the theory of consortia has gained more and more ground, in which the presence of patterns of linked microbes has a decisive biological function. Therefore, the study of the effect, if any, of a food intervention on the microbiota should consider the latter as a set of populations of interconnected consortia with compensatory capacities to adapt to the different nutritional intake.

As an example, shotgun metagenomic sequencing of 595 faecal samples showed that a certain lactose intolerant population genotype can balance the lower host’s β-glucosidases expression due to a greater abundance of bacterial taxa expressing β-glucosidases [8]. To study the consortium nature of the microbiome, we must rely on specially designed data analysis tools of different origins, metagenomic or metamolecular, capable of evaluating the existence of interconnections between populations of microbial individuals. The term metamolecular, in this context, is used to refer to those molecules that are generated by consortia in which some species can utilise, as a carbon source, substrates generated by other species, releasing otherwise inaccessible metabolites.

The intertwined nature of these omics has been studied as such in recent years. One of the first works investigating metabolome–microbiome crosstalk with correlation maps and networks was by Mc Hardy et al. [9]. This work set a framework to investigate the metabolome (mass spectrometry) and microbiome (OTUs) from cecum and sigmoid colon tissues. To find dependencies between microbes and gut epithelial metabolites, they used clustering and correlation maps of features to build a network. The resulting network is, however, characterised by simple measures (e.g., total number of links, average number of links per type of features) without taking its topology into account. Recently, another work [10] studied the links between gut microbiome and stool metabolome profiles in patient with lung cancer using the WCGNA framework [11]. The resulting correlation networks are characterised by topological measures such as the betweenness centrality and clustering coefficient. This allowed for a study focused on modularity and communal structures of the networks, without, however, taking communal overlap into account.

To this aim, omics data from faecal microbial taxa, stool metabolome (1H NMR spectra), and stool volatilome (GC-MS) were integrated to achieve an efficient crosstalk between inter-omics features (Figure 1). For visualisation and methodological discussion purposes, networks were generated from datasets obtained from faecal samples coming from the nutritional intervention trial performed within the framework of the EU-funded project PATHWAY-27 (grant agreement n° 311876).

## 2. Methods

In this section, the methodologies applied to obtain data used to test and visualise the proposed framework are also described (Section 2.1 and Section 2.2). The intervention trial providing the stool samples used to generate the datasets was a randomised, parallel, double-blind, placebo-controlled intervention trial including 325 participants. These characteristics are considered the gold standard for the network’s construction. Participants were randomly divided into 4 groups, each 1 receiving: (1) placebo; (2) dairy food enriched with docosahexaenoic acid (DHA) plus beta-glucan; (3) egg-based food enriched with DHA plus anthocyanins (AC); or (4) bakery food enriched with DHA plus AC. The intervention study lasted 12 weeks. Volunteers were asked to participate in a sub-study on a voluntary basis. In the sub-study, stool samples were collected in the beginning and at the end of the trial. For data analysis, only volunteers that completed the study with a compliance to the treatment >70% and provided stool samples at the beginning and end of the trial were considered (N = 90). Further details of the intervention study are reported in [12]. The following methods are presented as standard operating procedures.

### 2.1. Sampling and Data Collection

Material Stool samples, once collected, are aliquoted in duplicate at baseline (T_0_) and at the end of the trial (T_e_), and are labelled as the test group (indicated as A diet) and the placebo (indicated as P diet).

For the sequencing data analysis, the QIIME pipeline version 1.5.0 [13] is used. Within-community diversity (alpha-diversity) is calculated using observed OTUs, and the Chao1, Shannon, Simpson, and Good’s coverage indices with 10 sampling repetitions at each sampling depth. Student’s *t*-test is applied to compare the latest sequence/sample values of different treatments within an index. An analysis of similarity (ANOSIM) and the ADONIS test are used to determine statistical differences between samples (beta-diversity) following the QIIME compare_categories.py script by using weighted and unweighted phylogenetic UniFrac distance matrices. Principal coordinate analysis (PCoA) plots are generated using the QIIME beta-diversity plots workflow [14]. For the volatilome dataset, normality is checked with the Shapiro–Wilk test, whereas homoscedasticity is evaluated with Levene’s test [15]. Differences between all samples are evaluated with an untargeted analysis of variance (ANOVA) set at *p* < 0.05. Additionally, for the data analyses, Statistica version 8.0 (Tibco Inc., Palo Alto, CA, USA) is used. Normalisation of datasets is performed with the mean centring method. The outliers’ values that stand out from the pattern of the values in the dataset are discarded.

### 2.2. Microbiota Characterisation

#### 2.2.1. DNA Extraction, Amplification, and Sequencing

Stool samples are received frozen from the recruitment centre and kept at −80 °C for short-term storage. DNA is extracted from stool samples using the Purelink Microbiome DNA Purification Kit (Invitrogen, Thermo Fisher Scientific, Carlsbad, CA, USA). Nucleic acid purity is tested on the BioDrop Spectrophotometer (Biochrom Ltd., Cambridge, UK). The baseline and the endpoint are used for MiSeq sequencing (Illumina Inc, San Diego, CA, USA). Considering the MiSeq approach, bacterial diversity is obtained through library preparation and sequencing of the 16S rRNA gene. The following two amplification steps are performed: an initial PCR amplification using 16S (V3-V4) locus-specific PCR primers (16S-341F 5′-CCTACGGGNGGCWGCAG-3′ and 16S-805R 5′-GACTACHVGGG TATCTAATCC-3′) and a subsequent amplification integrating relevant flow-cell binding domains (5′-TCGTCG GCAGCGTCAGATGTGTATAAGAGACAG-3′ for the forward primer and 5′-GTCTCGTGGGCTCGGAGATGTG TATAAGAGACAG-3′ for the reverse overhang); lastly, unique indices are selected among those available.

Nextera XT Index Kits are used according to manufacturer’s instructions (Illumina Inc., San Diego, CA, USA). Both input and final libraries are quantified with the Qubit 2.0 Fluorometer (Invitrogen, Thermo Fisher Scientific, Carlsbad, CA, USA). In addition, the libraries are quality-tested by the Agilent 2100 Bioanalyzer High Sensitivity DNA assay (Agilent technologies, Santa Clara, CA, USA). Libraries are sequenced in an MiSeq (Illumina Inc, San Diego, CA, USA) at the paired ends with 300 bp read length [14]. Reads are demultiplexed based on the Illumina indexing system. Sequences are analysed using QIIME 1.5.0 [13]. After filtering based on read quality and length (minimum quality = 25 and minimum length = 200), operational taxonomic units (OTUs) defined by a 97% similarity are picked using the Uclust v1.2.22 q method [16], and the representative sequences are submitted to the RDP classifier [17] to obtain the taxonomy assignment and the relative abundance (RA) of each OTU using the Greengenes 16S rRNA gene database [18]. Alpha- and beta-diversity analyses are performed using QIIME 1.5.0 and used as a quality control for the microbiome dataset.

#### 2.2.2. Gas Chromatography-Mass Spectrometry for Volatilome

Experimental data of volatile metabolomic profiles in faeces are obtained by solid-phase microextraction–gas chromatography–mass spectrometry (SPME–GC–MS) analysis. For the volatilome analyses, the same samples used for microbiota analyses are extracted. Volatile organic compound (VOC) evaluation is carried out on an Agilent 7890A Gas Chromatograph (Agilent Technologies, Santa Clara, CA, USA) coupled to an Agilent Technologies 5975 mass spectrometer operating in the electron impact mode (ionisation voltage of 70 eV) equipped with a Chrompack CP-Wax 52 CB capillary column (50 m length, 0.32 mm ID) (Chrompack, Middelburg, The Netherlands). The solid-phase microextraction (SPME)–GC–MS protocol is followed and the identification of volatile compounds is performed according to previous reports, with minor modifications [19,20,21,22,23].

Briefly, 3 mL of vessel content or faecal slurry is placed into 10 mL glass vials and added to 10 μL of the internal standard (4-methyl-2-pentanol) to reach a final concentration of 4 mg/L. Samples are then equilibrated for 10 min at 45 °C. SPME fibre, coated with carboxen-polydimethylsiloxane (85 μm), is exposed to each sample for 40 min. Preconditioning, absorption, and desorption phases of SPME–GC analysis and all data-processing procedures are carried out according to previous publications [21,22,23]. Briefly, before each headspace sampling, the fibre is exposed to the GC inlet for 10 min for thermal desorption at 250 °C in a blank sample. The samples are then equilibrated for 10 min at 40 °C. The SPME fibre is exposed to each sample for 40 min, and finally, the fibre is inserted into the injection port of the GC for a 10 min sample desorption. The temperature program is: 50 °C for 1 min, then programmed at 1.5 °C/min to 65 °C, and finally, at 3.5 °C/min to 220 °C, which is maintained for 25 min. Injector, interface, and ion source temperatures are 250, 250, and 230 °C, respectively. Injections are carried out in splitless mode and helium (3 mL/min) is used as a carrier gas. Identification of molecules is carried out by searching mass spectra in the available databases (NIST 11 MSMS library and the NIST MS Search program 2.0 (NIST, Gaithersburg, MD, USA)).

The relative abundance of volatilome data is expressed as a percentage of the peak area with respect to the total area of the chromatograms, and, as quality control, some stool-associated compounds are absolutely quantified in mg/kg (LOQ = 0.03 mg/kg and LOD = 0.01 mg/kg) [24]. For these latter compounds, samples at the endpoint are compared to the baseline and values are expressed as shifts. All results are expressed as normalised mean values obtained from technical duplicates for two independent biological replicates.

#### 2.2.3. 1H Nuclear Magnetic Resonance Spectrometry (Untargeted)

Faecal samples are prepared for nuclear magnetic resonance (NMR) analysis by vortex mixing for 5 min 80 mg of stool with 1 mL of deionised water, followed by centrifugation for 10 min at 14,000 rpm and 4 °C. About 540 µL of supernatant is added to 60 μL of a D2O 1.5 M phosphate-buffered solution containing 0.1% TSP (3-(trimethylsilyl) propionic acid-d4), NaN_3_ 2 mM, set at pH 7.4. Before analysis, samples are centrifuged for 10 min again, and then, 590 µL is transferred into an NMR tube.

Here we define the NMR parameters. Proton NMR (1H NMR) spectra are recorded at 298 K with an AVANCE III spectrometer (Bruker, Milan, Italy) operating at a frequency of 600.13 MHz. The hydrogen deuterium oxide (HOD) residual signal is suppressed by presaturation, whereas broad signals from slowly tumbling molecules are removed by including a Carr–Purcell–Meiboom–Gill filter to a free induction decay sequence. The filter is made of a train of 400 echoes separated by 800 μs, for a total time of 328 ms. Each spectrum is acquired by summing up 256 transients using 32 K datapoints over a 7211.54 Hz spectrum (for an acquisition time of 2.27 s). The recycle delay is set to 8 s, keeping in consideration the longitudinal relaxation time of the protons under investigation.

Spectra are processed with TopSpin 3.0 (Bruker) by using an automatic command apk0.noe, which performs the baseline and phase correction in one shot, and by applying a line-broadening factor of 1 Hz. The peaks are assigned by comparing their chemical shift and multiplicity with the literature [25,26] and by using Chenomx NMR suite 8.1 software. An in-depth view of the critical aspects related to NMR-based metabolomics could be appreciated in a pragmatic guide to metabolomics methodologies [27].

Here we define 1H NMR spectra preprocessing. After Fourier transformation, and phase and baseline correction, spectra are calibrated with reference to the chemical shift of 0.0 ppm assigned to the internal standard TSP; spectral peripheral regions together with the water signal are removed. After this, spectra are normalised, employing the probabilistic quotient algorithm (PQN) [28] on three different regions separately, namely, the aromatic, hydroxylic, and aliphatic regions (regional scaling), since this works best for these types of samples. After normalisation and prior to any possible statistical analysis, spectra are binned into intervals of 150 datapoints. As a result, the new spectral profile consists of 400 binned data and is saved as a text file.

### 2.3. Merging Omics Datasets

#### 2.3.1. Microbiome Data Processing

To delete features with distributions that may cause spurious effects in feature importance evaluation with machine learning methods, the relative abundances of taxa at T_0_ (baseline) and T_e_ (endpoint) are filtered and OTUs with a median abundance <0.5% are excluded. The remaining taxa abundances, which are characterised by skewed distributions approaching a log-normal, are log-transformed to approximately conform to normality. This is a well-known routine to reduce skewness and ensure that features are suitable for methods that assume normality. All microbial features at species level are kept for the merged dataset. Note that the 0.5% relative abundance threshold is a very hard one, which is chosen to generate a relatively small final network of integrated features for visualisation purposes and to facilitate a clearer appreciation of the proposed methodology.

#### 2.3.2. Volatilome Data Processing

A similar approach is followed for volatilome features obtained from gas chromatography/mass spectrometry. Relative abundances (RAs) of compounds are filtered at T_0_ and T_e_; therefore, compounds with a relative abundance inferior to 0.5% are filtered out. The remaining compounds, once again characterised by skewed distributions approaching a log-normal, are log-transformed to approximately conform to normality. The selected compounds exceeding an RA >0.5% are kept for the merged dataset.

#### 2.3.3. 1H NMR Feature Selection and Agglomeration

To reduce the number of spectral features and build a final merged dataset with a lower risk of redundancy and spurious spectral features, we propose a two-step feature selection. To delete completely uninformative features with respect to the ones linked to the effect of diets, a self-optimising multi-class logistic regression with L1 penalty is trained using the scikit-learn package (scikit-learn: machine learning in Python [29]) in a Python 3.8 environment. The L1 penalty regularises the coefficient of the multivariate logistic regression, thus features completely uncorrelated with outcomes (the different food intervention followed by subjects) are assigned a coefficient of 0. In other words, sparsity in the coefficient matrix is induced by the regularisation.

After this feature selection, approximately one third of the original spectral buckets along the entire ppm range of the spectra are kept as informative features. To avoid the risk of still having many correlated spectral features due to adjacent spectral buckets belonging to the same signals or different signals belonging to the same molecule, and to summarise peaks in the spectra belonging to correlated molecules, a feature agglomeration is introduced. A correlation-based hierarchical clustering is trained to group highly correlated features using the sklearn.cluster module of the scikit-learn package in a Python 3.8 environment. The dendrogram is then pruned so that features belonging to the same cluster has a maximal cophenetic distance of 0.7. This is the minimum cophenetic distance for which no singleton clusters (clusters with a single member) appear, and the number of resulting agglomerates is comparable with the number of features from other omics in the dataset, making for a good threshold that avoids an excessively fine partition of the spectral features and the formation of giant, uninformative agglomerations.

After this step, each agglomeration of features is summarised using the median of the spectral buckets in the cluster as a pooling function. The result is a limited number of spectral agglomerates kept for the merged dataset at T_0_ and T_e_. As shown in the next section, this two-step feature selection and transformation is crucial for the final network construction. In terms of network topology, this procedure ensures the construction of a merged network without a heavily connected giant component of spectral features. In other terms, a giant component of spectral features with a high density of links between them would only reflect, for the most part, the expected high correlation between spectral buckets corresponding to signals of the same molecule. This trivial and uninformative characteristic of spectral features would, in turn, affect the global network structure, making the emergence of interesting properties of the networks difficult to study when investigating topology.

#### 2.3.4. Final Merged Dataset

The dataset used to build up the examples presented in the methodological discussion, resulting from the choice of processing parameters and feature selection, consists of:Forty-two log-transformed relative abundances of microbial taxa;Log-transformed relative quantifications of fourteen volatile compounds;Thirty-four spectral agglomerates summarising metabolomic profiles. These agglomerates represent pooled sets of highly correlated spectral buckets, containing intramolecular resonances or resonances of interdependent molecules.

The same features are compared at T_0_ and T_e_ for each sex and treatment group.

### 2.4. Network Construction

For the purposes of the paper, which has a methodologic focus, the networks built from the merged datasets at T_0_ and T_e_ are undirected and unweighted to propose a scenario as simple as possible. To evaluate the effect of diets also considering gender-related differences, separate networks are generated for males and females at T_0_ and T_e_. At T_0_, each network is generated using the whole group of females or male subjects, regardless of the following intervention (A and P). A link between two nodes (the different types of features: volatilome, microbiome, or stool metabolome) exists if a Spearman correlation greater than 0.5 between the features is detected. At Te, each network is generated using subjects separated by sex and nutritional intervention (A or P). A link between two nodes exists if a Spearman correlation greater than 0.7 between the features is detected. Indeed, since the statistical significance of a correlation is a function of the sample size for which it is calculated, the higher sample size at T_0_ (A + P in each gender) than at T_e_ accounts for a softer threshold. At Te, when networks are generated considering both gender and dietary intervention (A or P), a harder threshold is needed to avoid representing unsignificant correlations.

In general, it could be a viable choice to set the minimum correlation threshold to the minimum value that yields a significance *p*-value <0.05 for the correlation, whose test statistics are basically a function of the size of the sample used to estimate such correlations. In this way, we are immediately guaranteed to build a network with meaningful links and have a threshold value that increases to detect sets of strongly connected nodes in the network. Another criterion is given by the soft-thresholding method that is present in correlation network analysis packages such as WCGNA [11]. The method consists of elevating correlations to the power that induces the maximum topological overlap with a scale-free topology for the network. This method is useful when assuming a scale-free topology is a necessity.

The use of the Spearman correlation allows us to catch nonlinear relationships between features, as a high Spearman correlation coefficient exists when a general monotonous relationship function is present between two features [30]. To build the network as unweighted and undirected, the resulting adjacency matrix is a Nfeats X Nfeats binary matrix of 0s and 1s, where 0 denotes a nonexisting link between two features due to the correlation being lower than the specified threshold and 1 denotes an existing link between two features due to the correlation being higher than the specified threshold. Rendering, subnetwork creation, and topologic analysis is performed using Cytoscape 3.7.2 [31].

## 3. Results and Discussion

From a multiomics perspective, building a correlation network from features of various natures measured in a group of subjects can be seen as a way to grasp a complex characterisation of that group, with the possibility of studying this characterisation at many biologically relevant levels. To understand this, it is useful to focus on how the networks proposed in this work are built and the meaning of a link appearing between two nodes, which is a representation of the features in the dataset. A link exists between two nodes when a given Spearman correlation threshold weight is satisfied, meaning that a monotonous relationship exists between two features that can be of the same type or of different types. This means that the generated binary network is a snapshot of all the relationships existing amongst features.

Links among features are undirected, meaning they have no direction and just represent an existing relationship without a privileged orientation, and are unweighted since the intensity of the relationship between two nodes is not measured and links just exist or not. Exploring this ensemble of relationships, which derive from the existence of a monotonous function of correlation amongst measures in the dataset, means exploring biologically relevant mechanisms that characterise the group for which the network is built (e.g., a heavily connected group of microbial species with a metabolite that is known to be a by-product of their metabolism appearing in a certain group).

Consequently, networks of features generated from two different groups can be compared in terms of the density of their links (how many nodes are connected to each other in each network) and the topology of the links (which nodes are connected in a group but not in the others, and where highly connected parts of the network arise, if any exist). When networks of different groups are generated with the same correlation threshold from the same features, a difference in link density and connectivity may signify a difference in detectability of certain patterns and mechanisms, thus somehow characterising the group.

Since the extent to which relationships are represented is dependent on the chosen correlation threshold, a low threshold allows for a representation of the system with an overall higher complexity, visualising weaker relationships between the features and generating a more globally interconnected network. Opposite of this, a high threshold conveys a more fragmented representation of the system, where small but strongly connected components are more likely to be represented (Figure 2).

Varying the correlation threshold weight when generating a network, depending on the diversity of the dataset, can be considered similar to zooming in and out with a microscope when observing something like a composite material; at each level of resolution, different structural properties are bound to emerge. Once the networks are built, the following step is the interpretation of their global organisation. For large networks, a well-known method is based on verifying if the node-degree distribution approaches a power law [32].

Briefly, if the characteristic quantities of a network (node-degree distribution, clustering coefficient distribution, etc.) follow a power law, the network is scale-free, and it can be assumed that it is generated through preferential attachment, i.e., there is a high probability that new nodes attach themselves to nodes with an existing high number of links. This mechanism gives rise to a network with a certain number of nodes with a degree (number of links) greatly exceeding the average that serve as hubs. The study of these hubs, their neighbourhood, and which purpose they serve in the network should give information about the overall structure and functionality of the network. An example of a comparison of networks from different groups in the dataset, using these types of statistics, is reported in Figure 3. This is an example of very different networks from a topological point of view, possibly reflecting the effects of the different nutritional intervention followed by the two groups.

However, the ideal behaviour of the scale-free properties does not happen as frequently for smaller natural networks as the networks of omics features that are generated in this kind of experiments, and it is difficult to infer certain properties of a network when it is outside the scale-free assumption, let alone compare different networks and ascertain biological characteristics from topological properties. A much more common feature of not-so-large networks is the presence of many parts, in which nodes are more highly connected to each other than to the rest of the network. These sets of nodes are called clusters, groups, communities, or modules depending on how their presence is detected in mathematical terms.

In general, the recurrence of a communal structure reflects the hierarchical nature of complex systems and is an interesting perspective to study patterns of omics features. Communities of different features and overlapping features connecting different communities are the important pieces of the biochemical snapshot seen through different omics. In this study, we define a community according to the previous literature [33,34], and also use the k-clique community finder algorithm proposed by the latter. The core idea of the definition of a community is that a typical community consists of several complete (fully connected) subnetworks with a tendency to share many of their nodes.

Rigorously, a k-clique community is a union of the complete subnetworks of size k (k-cliques), that can be reached from each other through adjacent k-cliques sharing k-1 nodes [35]. This definition encases two key concepts that can be translated to the representation of the correlation of omics features of different types: the existence of a structure of overlapping and nested communities, and the fact that members of the same community can be reached from each other through subsets of well-connected nodes. Studying the overlap and nesting of groups of multiomics features, which by definition and construction are linked as a function of their Spearman correlation, can result in the identification of multiomics signatures for the group from which the network is generated.

We can identify which well-connected subgraphs exist in the network at a given order k that serve as a sort of “zooming” parameter and speculate about which group-defining biochemical mechanisms we are observing in terms of metabolome, volatilome, and microbiome. Since nodes belonging to more than one community can be considered, by definition, important elements of the network, and each community is an ensemble of adjacent cliques, biochemical mechanisms underlying the interactions that constitute a community can be linked in a hierarchical way.

As an example, if two communities, each one formed by highly connected nodes representing different types of omics features, share a single node, through the shared node we can look at each single mechanism separately and in great detail. At the same time, we can link the single mechanisms as parts of a super-mechanism, i.e., each community represents by-products of a metabolic pathway that are shared by different microbial species and can be identified as parts of a super-pathway through the shared node. Furthermore, exploring the structures of the communities around overlapping nodes, and comparing which nodes are emerging as bridge nodes between communities in different treatment groups, allows for exploring the effects of the nutritional intervention. In Figure 4, an example of overlapping k-clique communities of order three found in the network generated by the group of female subjects that underwent the food intervention is reported.

## 4. Conclusions

In this work, a novel path for multiomics studies from data sampling to the construction of networks of integrated features was proposed. The case study proposed to discuss the methodology involved faecal microbial taxa, stool metabolome (1H NMR spectra), and GC-MS for stool volatilome. Particular attention must be paid when dealing with 1H NMR spectral features. To avoid hindering the quality of information in the networks, which can be impacted by the presence of many correlated parts of the NMR spectra, we introduced a two-step dimensionality reduction method. Filtering, transforming, and selecting features from the different omics ensures the building of optimal networks for exploration.

Links in the networks were generated using a Spearman correlation threshold amongst the features, thus a link between two features (intra-omics and inter-omics alike) exists if a monotonous relationship exists between them. Building a network of correlations at baseline and at the end of treatments allows us to investigate the effect of a certain food intervention as a web of complex interactions rather than variations of single features of different types. Focusing on detecting the communal structure appearing in the networks can be a good strategy to characterise the effects of food interventions in a biochemically meaningful way. Indeed, we can detect densely connected groups of nodes, and translating this to the dataset is equivalent to finding interacting microbial communities and linking them to the metabolic by-products that characterise their environment. Furthermore, we can carry this out to different extents by changing the order of the communities we want to detect and imposing different correlation thresholds for link generation, such as zooming in or out with a microscope.

Another perk of the proposed methodology is the possibility of detecting nodes shared amongst communities to find groups of features belonging to super-pathways or super-niches, or interacting in a longitudinal way. This is opposed to traditional cluster maps, where the existence of an element belonging to multiple clusters is admitted only by hierarchical ordering. Finally, communal structures can be used to characterise the effects of food interventions. Baseline, placebo, and interventions can be compared in terms of the communal topology of their networks: the number of communities, their extension, the distribution of nodes shared amongst communities, the emergence of similar or dissimilar communities (communities containing or not containing the same nodes), and the emergence of similar communities but with different level of completeness (communities with the same nodes, but with missing elements in a group or another) are all interesting aspects to look at to define what is being impacted by the food intervention.

Although the main advantage of the method is the possibility of studying biochemical mechanisms and their physiological reflections and preserving their complexity, the shift of the descriptive paradigm required for this approach is not trivial. Methods applied to feature processing and network building require user-defined (or self-optimisable) parameters and thresholds, which must be applied after establishing a good rationale that depends on extensive knowledge of the dataset. Furthermore, knowledge of topological measures and operators of network analysis and their interpretation is required. However, this descriptive paradigm is necessary for a true multiomics integration of data, which is a powerful tool to investigate the complex physiological effects of nutritional interventions.

## Figures and Tables

**Figure 1 metabolites-12-00736-f001:**
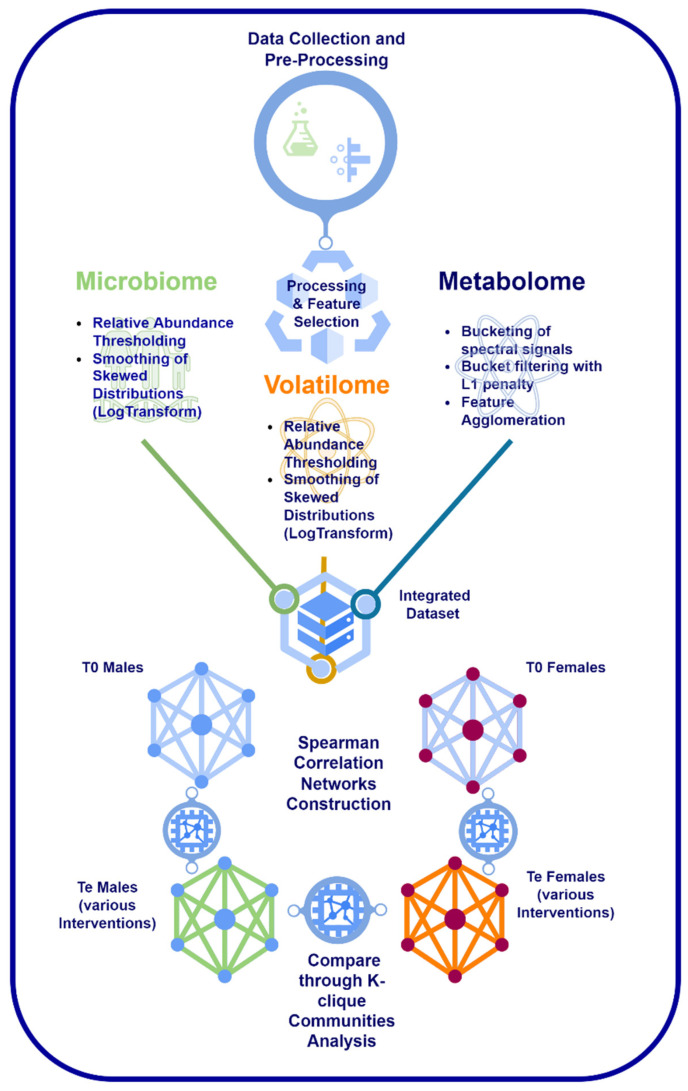
Visualisation of the omics networks: construction and analysis pipeline.

**Figure 2 metabolites-12-00736-f002:**
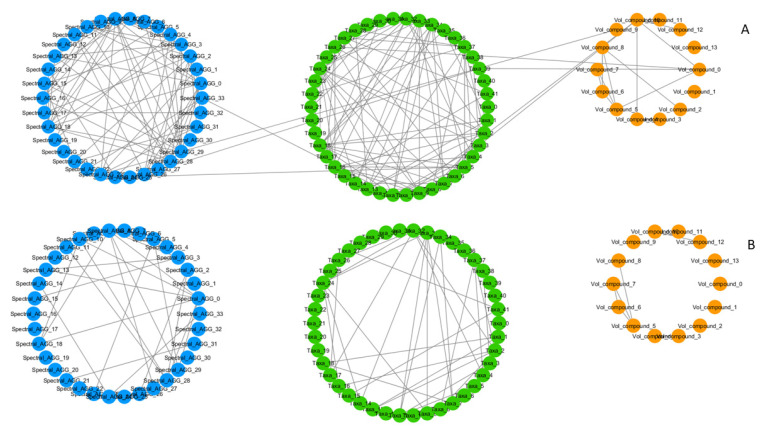
Correlation network of omics features generated for the same group (T_0_, males) at different correlation coefficient thresholds (t), with an attribute (feature)-grouped layout. (**A**) Softer correlation threshold, t = 0.5. High density of links in same feature-type groups and presence of inter-omics links (links connecting circular groups with different colours). (**B**) Harder correlation threshold, t = 0.6. Lower density of links in same feature-type groups, inter-omics links disappear. A more fragmented network of smaller but more strongly connected communities remains. Node colour represents different types of features. Blue: NMR spectral features, metabolome. Green: microbial species, microbiome. Orange: volatilome.

**Figure 3 metabolites-12-00736-f003:**
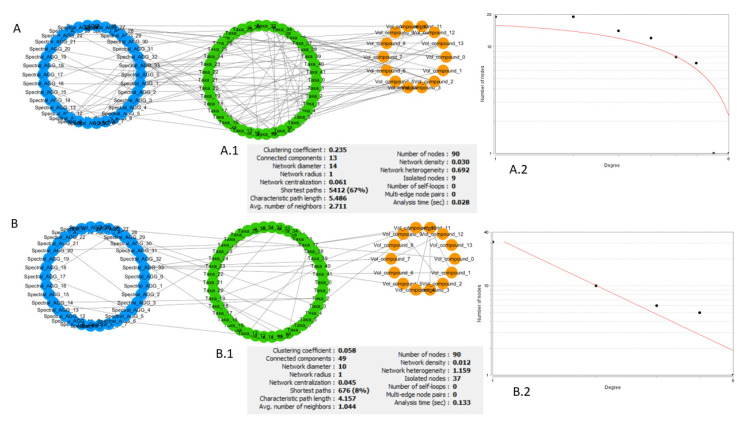
Comparison of network generated from different female diet groups at T_e_. (**A**) Network of omics interactions at T_e_ for females that underwent the food product intervention. (**A.1**) List of general parameters of the network. (**A.2**) Best fit of the node-degree distribution. Axes are set to be logarithmic so that the fit of a distribution approximating a power law is a line. (**B**) Network of omics interactions at T_e_ for females that underwent the placebo intervention. (**B.1**) List of general parameters of the network. (**B.2**) Best fit of the node-degree distribution. Axes are set to be logarithmic so that the fit of a distribution approximating a power law is a line. Comparing A.1 and B.1, we can see that network A is a denser network, with many interconnected neighbourhoods (avg. clustering coefficient) and lower number of isolated nodes. Its node-degree distribution (**A.2**) does not follow a power law and we cannot assume a scale-free structure and discuss its biological implications. Network B has a relatively high number of nodes with low degree (number of links) and a few nodes with very high degree (**B.2**). The node-degree distribution converges to a power law; the fit of the type y = ax^b^ returns R^2^ = 0.872. Node colour represents different types of features. Blue: NMR spectral features, metabolome. Green: microbial species, microbiome. Orange: volatilome.

**Figure 4 metabolites-12-00736-f004:**
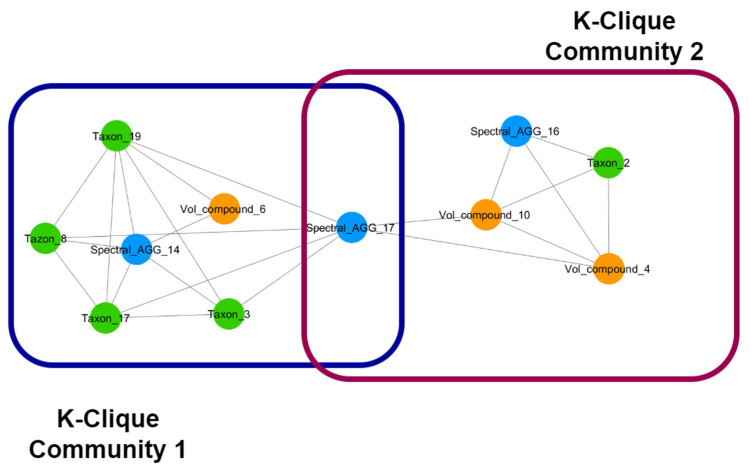
An example of the overlap of two clique communities of order three from a nutritional intervention in females. We can observe that the two communities are linked through the overlap of a node containing spectral features. The communities contain a mix of microbial species, volatile compounds, and spectral features that can identify the biochemical mechanism underlying the interactions amongst these features. Studying each community separately can help identify mechanisms at a higher detail; the fact that these two communities are linked through a shared node can help link the identified mechanisms in a hierarchical way. The distribution of nodes linking communities through overlap can also be used to characterise important aspects of such networks. Node colour represents different types of features. Blue: NMR spectral features, metabolome. Green: microbial species, microbiome. Orange: volatilome.

## Data Availability

The data presented in this study are available at https://github.com/CarloMengucci/Omic-Networks-Crosstalk (accessed on 26 June 2022).

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
