# Peer review of "K-Clique Multiomics Framework: A Novel Protocol to Decipher the Role of Gut Microbiota Communities in Nutritional Intervention Trials"

_metabolites, 2022, doi:10.3390/metabo12080736_

Round 1

Reviewer 1 Report

In this work, the authors attempt to build a framework for the multiomic analysis of gut microbiota in nutritional trials, and showed some network figures. However, without any inference to the biological significance or any interpretation of the dataset, it is somehow very difficult for the readers to grasp the significance of the work, and to compare it to other similar bioinformatic approaches in place. 

Author Response

Reviewer 1

In this work, the authors attempt to build a framework for the multiomic analysis of gut microbiota in nutritional trials, and showed some network figures. However, without any inference to the biological significance or any interpretation of the dataset, it is somehow very difficult for the readers to grasp the significance of the work, and to compare it to other similar bioinformatic approaches in place.

We thank the Reviewer for her/his very useful criticism.

To better explain the meaning of the manuscript, in the revised version we included a new figure, which should guide readers and improve their comprehension of the work. In the revised version, we have also made improvements on clarity and on details, and we have reported new sections to better readers’ comprehension, for example, a comparison of our framework to some existing in literature, with the explanation of what we propose is able to improve. Despite we appreciate the recommendation of the Reviewer about the opportunity to discuss biological significance and interpretation of data, we would like to keep the paper as much general as possible in order to be considered as methodological research that must be appreciated in terms of the logical pipeline bearing the newly proposed framework. Posing attention on the outcome, rather than on the pathway, could be misleading, as we could fall into two possible occurrences: a) the results are as expected based on the current knowledge; b) the results are not trivial and are obtained thanks to the novel approach. In the former case, the results are not necessary to grasp the power of the framework; in the latter case, the framework itself must be checked accurately for robustness. In this paper, we would like to focus the attention on the logical pipeline and on the ideal correctness of each single step, independently of the biological interpretation of the specific dataset. Rather, we give a hint about the biological relevance (as written in the discussion section and in the conclusions) of considering the networks of omic features to extrapolate the occurrence of microbial communities.

(x) English language and style are fine/minor spell check required

The MS was accurately revised accordingly

Reviewer 2 Report

In the current study the authors, in an effort to investigate the complex physiological effects of nutritional interventions, propose a novel path for multiomics studies. They construct networks of stool microbiome metabolome and volatilome and explore the inter-omics cross talk. The study is well written, the introduction clearly explains the aim of the study and the discussion convinces the reader about the importance of the method to explore overlapping communities. Although the study is very interesting, and fits in the scope of the journal, experts mainly can easily go through the whole procedure; thus supplementary material would probably be useful for the non experts.

Comment

-  The authors refer that the study was a randomized, parallel, double blind, placebo-controlled intervention trial including 200 subjects. Further information is needed

- Figures 2 and 3: add explanation of the colour of nodes     

Author Response

Reviewer 2

In the current study the authors, in an effort to investigate the complex physiological effects of nutritional interventions, propose a novel path for multiomics studies. They construct networks of stool microbiome metabolome and volatilome and explore the inter-omics cross talk. The study is well written, the introduction clearly explains the aim of the study and the discussion convinces the reader about the importance of the method to explore overlapping communities. Although the study is very interesting, and fits in the scope of the journal, experts mainly can easily go through the whole procedure; thus supplementary material would probably be useful for the non experts.

We thank the Reviewer for her/his appreciation and positive comments on our work. We totally agree that non-experts could have difficulties in going through the whole procedure described in the manuscript. As suggested by the Reviewer, a new figure was added to facilitate comprehension and the exemplificative architecture of the python pipeline is deposited in an open access platform (https://github.com/CarloMengucci/Omic-Networks-Crosstalk).

Comment

-  The authors refer that the study was a randomized, parallel, double blind, placebo-controlled intervention trial including 200 subjects. Further information is needed

We thank the Reviewer for the comment. In the revised version, further details of the study protocols were provided (lines 95-104). However, we would like to stress that the experiment used for the data collection is just an example of an experimental design suitable to be analysed with the proposed framework.

- Figures 2 and 3: add explanation of the colour of nodes   

We apologize and thank the Reviewer for noticing it. In the revised version, the explanation of the colour of nodes was included in the Figure captions. 

 (x) English language and style are fine/minor spell check required

The English language was accurately double-checked.

Reviewer 3 Report

The authors conducted a nutritional intervention study and collected the omics data on faecal microbial taxa, stool metabolome (1H NMR spectra) and GC-MS for stool volatilome. They aimed to integrate three omic data to elucidate an efficient crosstalk between omics features. The authors intended to build a network of correlations at baseline (T0) and end of treatments (Te) allow to investigate the effect of a certain food intervention as a web of complex interactions rather than variations of single features of different types. The networks built from the merged datasets at T0 and Te are undirected and unweighted. A link between two nodes (the different types of features: volatilome, microbiome or stool metabolome) exists if a Spearman correlation greater than 0.5 between the features is detected. They used different correlation threshold to zoon in and out the network detected. However, the entire study failed to provide novel insight into the impact of dietary intervention on the change of inter-omic features. It is a methodological paper missing many details.  They could do a better job to use the data to illustrate the method and feature selection, as well as to identify pathway that shows the connection or network of bacteria, metabolite, and volatile compounds. At the same time, the novelty of the approach should be highlighted if any. None of the important feature from any omic is reported.  

Some specific comments:

Figure 1, what is t = 0.6? Does it mean correlation coefficient?

The idea of zooming in and out using threshold is fine. Can the authors provide any insight on how to determine the optimal threshold using their data?

Line 337 – 342 may be shown in figure legend.

How many features are included in data analysis of each omic (each treatment group and T0 and Te)? Are the same types of features included when analyzing the data from T0 and Te, in male and female? Did the author include the standard for chemicals that aims to identify the important feature?  

Line 51 needs a coma.

Line 83 provide the exact sample size

Please provide the definition of outlier

Were T0 and Te sample analyzed in the same batch of experiments for microbiome and metabolomics? The quality controls measures should be presented. Lab variation can introduce a lot of noise.

If the manuscript intends to show the methodology of data collection, more details should be provided.

Which hypervariable region of 16s rRNA gene is sequenced?

In analyzing microbiome data, ASV-based method has better classification accuracy than the OTU-based method. OTU is spelled as OUT in several places.

For 1H Nuclear Magnetic Resonance Spectrometry and volatilome, what are the compounds measured? Is this targeted or untargeted method? What is the normalization and scaling procedure for metabolite data?

L203-L204 can be rephrased for clarity.

The author may use the past tense in writing.

Please define RA when first use. What is the relative abundance of volatilome data? Relative to what? What is the standard to identify the compounds?

Overall, the authors did not show how to use the data-driven multi-omic approach to decipher the effect of intervention. A few figures presented are not east to read and superficial. The proposed idea may be of interest, but it is pale without real data and real scenario. It did not provide the insight into how diet can impact gut microbiota and its metabolite. What are the important features of each omic? Do this omic cross-talk? Many important questions are not answered in this research that means to reveal the network of microbiome, metabolomics, and volatilomes in an intervention setting.   

Author Response

Reviewer 3

The authors conducted a nutritional intervention study and collected the omics data on faecal microbial taxa, stool metabolome (1H NMR spectra) and GC-MS for stool volatilome. They aimed to integrate three omic data to elucidate an efficient crosstalk between omics features. The authors intended to build a network of correlations at baseline (T0) and end of treatments (Te) allow to investigate the effect of a certain food intervention as a web of complex interactions rather than variations of single features of different types.

The networks built from the merged datasets at T0 and Te are undirected and unweighted. A link between two nodes (the different types of features: volatilome, microbiome or stool metabolome) exists if a Spearman correlation greater than 0.5 between the features is detected. They used different correlation threshold to zoon in and out the network detected. However, the entire study failed to provide novel insight into the impact of dietary intervention on the change of inter-omic features.

We thank the Reviewer for her/his comment, which allow us to better explain the aim of the research work. We understand the Reviewer concern. To dissipate it, the nature of the research should be considered. Indeed, the manuscript describes the protocol used to build up a bioinformatic tool to be used schematically free. To underline the versatility of application, we deliberately avoided giving experimental data. The aim of the study was not to provide novel insight into the impact of dietary intervention on the change of inter-omic features. In this paper, we would like to focus the attention on the logical pipeline and on the ideal correctness of each single step, independently of novel insights into the impact of dietary intervention. Rather, we give a hint about the biological relevance (as written in the discussion section and in the conclusions) of considering the networks of omic features to extrapolate the occurrence of microbial communities. We intend to propose a tool facilitating interpretation of multi-omic data from nutritional intervention.

It is a methodological paper missing many details. They could do a better job to use the data to illustrate the method and feature selection, as well as to identify pathway that shows the connection or network of bacteria, metabolite, and volatile compounds.

As requested, further details have been included in section 2.3.4 (lines 282-290). Data details are provided in sections 2.2.1, 2.2.2, 2.2.3. Features selection is described in sections 2.3.1, 2.3.2 and 2.3.3. To illustrate the general framework, we have included a new figure (Figure 1) describing the pipeline and the inputs of our protocol. Moreover, further details have been included in the figures’ captions.

At the same time, the novelty of the approach should be highlighted if any. None of the important feature from any omic is reported.  

In the revised version, we highlighted the novelty of the approach by comparing our work with other approaches in the literature (lines 70-83). Additional references were included to strengthen the novelty of the proposed methodology.

Some specific comments:

Figure 1, what is t = 0.6? Does it mean correlation coefficient?

We thank the Reviewer for the comments. “t” indicates the correlation coefficient threshold. In the revised version, we specified it in the caption of the newly numbered Figure 2.

The idea of zooming in and out using threshold is fine. Can the authors provide any insight on how to determine the optimal threshold using their data?

We are glad the Reviewer appreciates the idea of zooming. To help readers on how to determine the optimal threshold we reported this new paragraph at lines 307-313

In general, it could be a viable choice to set the minimum correlation threshold to the minimum value that yields a significance p-value < 0.05 for the correlation, whose test statistics is basically a function of the size of the sample used to estimate such correlations. In this way, we are immediately sure to build a network with meaningful links and have a threshold value to increase to detect sets of strongly connected nodes in the network. Another criterion is given by the soft-thresholding method that is present in correlation network analysis packages such as WCGNA [11].

Line 337 – 342 may be shown in figure legend.

We thank the Reviewer. The manuscript was revised accordingly

How many features are included in data analysis of each omic (each treatment group and T0 and Te)? Are the same types of features included when analyzing the data from T0 and Te, in male and female? Did the author include the standard for chemicals that aims to identify the important feature?  

We apologize for the missing information. In the revised version, we included a new paragraph to provide them (line 282-290).

Line 51 needs a coma.

Done, thank you.

Line 83 provide the exact sample size

Done, thank you.

Please provide the definition of outlier

The statistical definition of outlier is now reported (line 127-128) in the revised version

Were T0 and Te sample analyzed in the same batch of experiments for microbiome and metabolomics? The quality controls measures should be presented. Lab variation can introduce a lot of noise.

Samples were sent frozen to the University of Bologna from the different recruiting centers. They were kept frozen till the end of the trial. Thawing of samples was performed to analyse the same samples for microbiome and metabolomics in the same batch.

The quality controls we used were: i) for the Microbiome, Alpha and Beta Diversity as now reported in 2.2.1, ii) for the GCMS, absolute quantification of common stool VOCs as is now reported in 2.2.2, and iii) a specific reference as was reported in 2.2.3.

This information was included in the revised version of the manuscript (lines 156-157: 185-188)

If the manuscript intends to show the methodology of data collection, more details should be provided.

We agree with the Reviewer. Additional information of the methodology of data collection were included in the revised version of the manuscript (section 2.2). Moreover, an example of dataset for microbiome, volatilome and NMR spectral data are deposited in an open access platform (https://github.com/CarloMengucci/Omic-Networks-Crosstalk).

Which hypervariable region of 16s rRNA gene is sequenced?

The hypervariable region studied was V3-V4. This detail is now reported in the manuscript (line 139)

In analysing microbiome data, ASV-based method has better classification accuracy than the OTU-based method. OTU is spelled as OUT in several places.

Sorry, we disagree. We believe that the OTU style is the most common and user-friendly. OTU threshold and rarefaction use to have a major impact on ecological studies, especially biodiversity indices that can count on the quantification of rare species, in respect to only on presence/absence of units considered by the ASV. However, there are also major differences in the occurrence and relative abundance of major bacterial taxa identified by both approaches (Chiarello et al., 2022).

Chiarello, M., McCauley, M., Villéger, S., & Jackson, C. R. (2022). Ranking the biases: The choice of OTUs vs. ASVs in 16S rRNA amplicon data analysis has stronger effects on diversity measures than rarefaction and OTU identity threshold. PloS one17(2), e0264443.

We apologize for the misspelling. It was corrected in the revised version.

For 1H Nuclear Magnetic Resonance Spectrometry and volatilome, what are the compounds measured? Is this targeted or untargeted method? What is the normalization and scaling procedure for metabolite data?

We thank the Reviewer and apologize for the missing details. All NMR data analyses were untargeted. Details about scaling, normalisation and pre-processing procedures are reported in 2.2.3. For volatilome, methods are reported in 2.2.2.

what are the compounds measured?

For volatilome, the compounds measured are microbial and human metabolites, such as low molecular organic acids (Propanoic, Butanoic acids, etc), aldehydes (Hexanal, Benzaldehyde, etc), alcohols (Butanol, Propanol, etc,), aminoacids, simple sugars and others.

For NMR data, the untargeted analysis was applied directly on spectral features, further correlated as raw agglomerated data to the other volatilome and OTU data. It is well-known that NMR metabolomics is characterized by very high reproducibility but low resolution, thus making the exploitation of raw data better performing. Once the spectral features are emerging as relevant, then the subsequent signal assignment to specific compounds could be performed.

L203-L204 can be rephrased for clarity.

The sentence was revised for clarity as follows:

To delete features with distributions that may cause spurious effects in feature importance evaluation with machine learning methods, the relative abundances of taxa at T0 (baseline) and Te (end point) were filtered and OTUs with a median abundance < 0.5% were excluded.

The author may use the past tense in writing.

We did language corrections in the revised version and sometimes the verb tense was corrected, but we have also prepared the paper as a protocol, thus we retain the original tense also appropriate.

Please define RA when first use.

We apologize. The acronym explanation is now reported at first use (line 154).

What is the relative abundance of volatilome data? Relative to what?

The relative abundance of volatilome data is expressed as percentage of the peak area with respect to the total area of the chromatograms. We revised this part as follow (lines 185-191):

The relative abundance of volatilome data is expressed as percentage of the peak area with respect to the total area of the chromatograms, and, as quality control, some stool associated compounds were absolutely quantified in mg/kg (LOQ = 0.03 mg/kg and LOD = 0.01 mg/kg) [21-23]. For these latter compounds, samples at the endpoint (24 h) are compared to the baseline and values are expressed as shifts. All results are expressed as normalized mean values obtained from technical duplicates for two in-dependent biological replicates.

What is the standard to identify the compounds?

Compounds were identified based on comparison of their retention times with those of pure compounds (Sigma-Aldrich, Milan, Italy). Identification was confirmed by searching mass spectra in the available databases (NIST 11 MSMS library and the NIST MS Search program 2.0 (NIST, Gaithersburg, MD, USA) [lines 182-185]

Overall, the authors did not show how to use the data-driven multi-omic approach to decipher the effect of intervention.

We agree with the Reviewer that the overall procedure for applying the proposed framework to multi-omic data is missing. As suggested by the Reviewer, a new figure 1 was added to facilitate comprehension and the exemplificative architecture of the python pipeline is deposed in an open access platform (https://github.com/CarloMengucci/Omic-Networks-Crosstalk).

A few figures presented are not east to read and superficial.

We appreciate the Reviewer’s comment, and some details were included in a new Figure showing the pipeline of our approach, and the others were edited.

The proposed idea may be of interest, but it is pale without real data and real scenario. It did not provide the insight into how diet can impact gut microbiota and its metabolite. What are the important features of each omic? Do this omic cross-talk?

Real data in a real scenario were used as input of the proposed framework. However, as already clarified above, the topic of this paper is not the biological relevance of the results, but we intended to provide a description of the necessary dataset, opportunely framed, as well as the algorithms able to integrate and holistically analyse the network of relationships, to catch the insight into how the diet can impact on gut microbiota and how the latter could affect faecal metabolome. In this sense, all the omic features are relevant but only those related to an omics cross-talk will emerge case by case. Putting here emphasis on a specific clique would divert the attention from the issue we would pose: looking at the crosstalk of omics at various levels by studying the overlap of communities of omic features

Many important questions are not answered in this research that means to reveal the network of microbiome, metabolomics, and volatilomes in an intervention setting.   

The Reviewer is very clear and his/her last sentence sheds light on the misunderstanding: the paper is not presenting the results of an intervention study, but the outcome of an original research study aiming at providing a methodology.

(x) Moderate English changes required

The MS was accurately revised accordingly

Round 2

Reviewer 1 Report

The authors have addressed my concerns.

Author Response

We thank the Reviewer

Reviewer 3 Report

If the manuscript is a protocol, please add this clarification in the title to avoid the confusion as occurred to this reviewer. 

If it is a protocol, please provide enough details for each methodology so that other readers can repeat the work flow. 

Please try to avoid long paragraphs. For instance, Introduction can be divided to three to four paragraphs.  

Author Response

If the manuscript is a protocol, please add this clarification in the title to avoid the confusion as occurred to this reviewer.

According to the Reviewer’s suggestion, we have changed the title in:

K-clique multi-omics framework: a novel protocol to decipher the role of gut microbiota communities in nutritional intervention trials

If it is a protocol, please provide enough details for each methodology so that other readers can repeat the work flow.

In the revised version we provided all details for data acquisition (section 2), a link to the GitHub repository, where the python scripts are available, as well as the example dataset with a format suitable for the proposed framework. Moreover Figure 1 should give a general view of the protocol. This way, the reader could apply the whole protocol to their own data, provided that they have enough experience with python.

Please try to avoid long paragraphs. For instance, Introduction can be divided to three to four paragraphs. 

We have divided the different sections in paragraphs counting 120±40 words. This way, we could expect a better readability

This manuscript is a resubmission of an earlier submission. The following is a list of the peer review reports and author responses from that submission.